# Positive Effect Induced by Plasma Treatment of Seeds on the Agricultural Performance of Sunflower

**DOI:** 10.3390/plants12040794

**Published:** 2023-02-10

**Authors:** Ioana Florescu, Ioan Radu, Andrei Teodoru, Lorena Gurau, Constantina Chireceanu, Florin Bilea, Monica Magureanu

**Affiliations:** 1Research and Development Institute for Plant Protection, Bd. Ion Ionescu de la Brad 8, 013813 Bucharest, Romania; 2Department of Plasma Physics and Nuclear Fusion, National Institute for Lasers, Plasma and Radiation Physics, Atomistilor Str. 409, 077125 Magurele, Romania

**Keywords:** non-thermal plasma, seed treatment, dielectric barrier discharge, sunflower, crop yield

## Abstract

The need for efficient technologies to enhance productivity in agriculture strongly motivates research on plasma treatment of seeds and plants. In this study, the influence of plasma treatment on sunflower (*Helianthus annuus* L.) seeds was evidenced throughout the entire life span of the plants. The seeds were packed in a DBD reactor operated in air and treated in plasma for 10 min, using a sinusoidal voltage of 16 kV amplitude at 50 Hz frequency. Early growth observation of plants under laboratory conditions showed that, after a slower start, the plasma-treated seeds developed faster and produced taller seedlings with greater total mass as compared to the control samples. Results obtained from mature plants cultivated in the field revealed a positive effect of plasma exposure with respect to capitulum size, number of seeds per capitulum and mass per thousand seeds, resulting in a remarkable increase in crop yield. The plasma effect lasted for at least two weeks of seed storage; however, it was considerably affected by the sowing period.

## 1. Introduction

Plasma agriculture is a novel and rapidly developing research field, motivated on the one hand by the need for new and efficient technologies to enhance productivity in order to keep pace with the continuously rising demand for food, and, on the other hand, by the positive effects observed as a result of plasma treatment. Several recent reviews summarize the literature results from different perspectives, showing the ability of plasma to improve germination and enhance plant growth, to decontaminate seeds and to enhance stress tolerance and plants’ resistance to diseases. Attri et al. [1] mainly focus on the effect of seed treatment by non-thermal plasma (NTP) on germination and seedling growth, considering both low/medium-pressure plasma sources and atmospheric-pressure plasma, as well as indirect treatment by plasma-treated water (PTW). The possible mechanisms leading to the observed germination increase and growth enhancement are also briefly mentioned. The review of Scholtz et al. [2] is dedicated to the microbicidal effect of NTP and confirmed that surface contamination of various cereal grains caused by bacteria and fungi can be reduced/eliminated by plasma treatment, without deterioration of the grains or decline in the cereal quality. An overview of the plasma agriculture literature in the context of the requirements for plant growth at various stages of plant development is provided by Ranieri et al. [3]. The authors conclude that the acceleration of germination produced by plasma treatment of seeds is most likely due to the combination of higher water uptake—caused by modifications to the seed coat that increase hydrophilicity—and the inhibition of dormancy initiated by plasma-generated reactive oxygen species (ROS). Similarly, irrigation with PTW may affect the chemical composition of the seed coat and may trigger changes in the hormone balance, also promoting germination. The role of nitrogen in plant growth as well as the effects of ROS are also addressed in the literature. Controlled delivery of plasma-generated species, adapted to plant needs at different development stages, is considered to be the key strategy for successful application of plasma in the agriculture field. The prospect of using plasma to stimulate the expression of various genes related to embryonic development, plant growth, resistance to pathogens and tolerance to abiotic stress is emphasized in the review of Leti et al. [4]. Novel findings evidencing changes induced by NTP treatment of seeds at the molecular level are analyzed in detail by Mildaziene et al. [5]. The authors provide a comprehensive overview of existing knowledge of the impact of plasma on biochemical and physiological processes in plants. Changes triggered by plasma in gene and protein expression, and in enzyme activities are considered, revealing the activation of stress response pathways, regulation of cell division and growth, and enhanced secondary plant metabolism. Effects on the amount of phytohormones and on hormone balance are discussed, as well as the stimulation of photosynthesis, enhancement of metabolic activity, and adaptive plant responses to abiotic and biotic stressors, leading to improved tolerance and stress resistance.

Many plant species have been investigated in connection with plasma treatment. Many of these studies were performed on cereal grains, due to their importance as a basic food, but several also deal with vegetables and fruits, legumes, etc. Only a few publications report results on the plasma treatment of sunflower [6,7,8,9,10,11,12], which is a valuable agricultural crop, cultivated mainly to produce vegetable oil, but also used for livestock and bird feed, human consumption, the production of biofuel and medicinal purposes. Some of the studies on NTP treatment of sunflower seeds focused on the influence of the treatment on germination and seedling growth under controlled conditions only over a few days [6,7]. Matra et al. [6] treated the seeds with a multi-needle positive DC corona discharge, operated in Ar-O_2_ mixture at voltages in the range of 8–14 kV, for durations of 1, 3 and 5 min. The plasma effect was evaluated by measuring the dry weight and shoot length of the seedlings after seven days of cultivation in the laboratory. Significantly longer and heavier sprouts were obtained from the seeds exposed to plasma under most experimental conditions, but without a clear dependence on the applied voltage nor treatment time. Sarapirom et al. [7] compared three types of plasma reactors for sunflower treatment: a low-pressure RF inductively coupled discharge, a surface dielectric barrier discharge (SDBD) and a plasma jet. The results (length and weight of sprouts) obtained with a certain plasma source seem to depend on the experimental conditions (i.e., discharge power and treatment time); in some cases the plasma effect being beneficial, while in others, detrimental. The authors stated that the SDBD was the most efficient, and indeed, at first sight it produced the longest seedlings; however, the results of the control samples also varied widely, so the data is actually inconclusive.

In other studies, a correlation was attempted between the morphometric parameters and changes in the phytohormone content and balance, and the gene expression [9,10,11], in order to elucidate the plasma mechanism of action. It was found that the seeds rapidly respond to plasma treatment on the level of phytohormone balance. Exposure to a DBD plasma operated in air at atmospheric pressure and at relatively high power density (3.05 W/cm^2^) induced significant changes in the amount of seed phytohormones that could be correlated with seedling growth, photosynthetic activity and, to a lesser extent, germination kinetics [9]. Thus, analyzing the two hormones that antagonistically regulate plant development (gibberellin—GA and abscisic acid—ABA), it was revealed that the positive effects corresponded to an increased GA/ABA ratio, while the negative impact on growth was related to decreased GA/ABA. A substantial shift in the phytohormone balance was also detected by Mildažienė et al. [10] as a consequence of seed exposure to a low-pressure RF capacitively coupled discharge, as well as after electromagnetic-field treatment, and simply as a result of keeping the seeds in vacuum. The authors indicated that each type of treatment generated a specific response: plasma treatment considerably increased the GA content, the electromagnetic field produced a decline in ABA, while vacuum affected the auxin/cytokinin balance. A stimulating effect on the expression of proteins involved in photosynthesis, and in processes regulating the photosynthetic activity, was also observed. Han et al. [11] investigated the effect of the treatment of sunflower seeds in an AC atmospheric-pressure DBD plasma, operated in argon at 16.8 kV voltage amplitude and 60 Hz frequency for short treatment times, up to 30 s. It was found that plasma treatment accelerated seed germination and promoted growth of sunflower seedlings, while molecular analysis on 14-day-old seedlings revealed augmented concentrations of soluble proteins in leaves, higher antioxidant enzyme activity and increased expression of growth-regulating factors in sunflower leaves, suggesting a mechanism based on enhanced energy metabolism and related gene expression.

Only one study reported results on mature plants grown under field conditions [12] and described the impact of plasma on stimulating root and lateral organ growth, which was attributed to the modification of the plant-associated microbial composition as a result of plasma treatment.

The present work investigates the effect of the treatment of sunflower seeds by a DBD plasma in air and monitors the plants grown from plasma-treated seeds throughout their entire lifespan. The plant development was followed both under laboratory conditions and in the field.

## 2. Results and Discussion

The germination of seeds in Petri dishes on the 4th day was the same (98%) for the untreated and plasma-treated samples. The radicles of the plasma-treated seeds were approximately 10% shorter as compared to the control seeds, and this difference is statistically significant. However, once transferred into soil, the plants grown from plasma-treated seeds developed faster and soon became taller than those which were untreated. Figure 1a shows the stem length (from the soil up to the cotyledons) of the treated and control plants cultivated in organic substrate and under controlled conditions over two weeks of observation. In the beginning of this period, the length was similar for the two samples, but further on, statistically significant differences in favor of the seeds exposed to plasma became visible. At the end of the observation interval, i.e., after 30 days of incubation, the mean shoot length of plasma-treated plants was significantly larger than for the untreated ones, while the values of the root lengths were similar (Figure 1b). The cotyledons’ dimensions, measured on the 10th day after treatment, were unaffected by plasma. Figure 1c,d show the fresh and dry weight of the plants on the 30th day, respectively. Plasma treatment enhanced biomass accumulation by approximately 30% (fresh weight) and 20% (dry weight). The differences from the control plants were, in general, statistically significant, with the exception of the dry weight of plant roots, which remained almost unchanged.

For the field trials, the seeds were sown on three dates approximately two weeks apart (12 April 2021, 26 April 2021, 7 May 2021), depending on the climatic conditions in the experimental area, especially precipitation. It is known that sowing time is one of the key factors affecting sunflower production. Early sowing permits sunflower plants to exploit the soil humidity and to avoid the period of high temperature and water stress during the flowering and seed-filling phases [13]. Early sowing also allows the plants a longer growing and developing season. In order to counteract the shortcoming during early sowing caused by the soil temperature being lower than optimum, stimulating the seeds by plasma treatment can be an alternative to be considered. Measurements made during plant growth, about 2–2.5 months after sowing, are presented in Figure 2.

The results revealed the dependence of the plants’ height, capitulum diameter and number of leaves, on the sowing date. For the optimum sowing period (i.e., April), the seeds exposed to plasma produced significantly taller plants, with more leaves and larger capitulum. The difference as compared to the control was especially significant for the earliest sowing date. Thus, for the seeds sown on 12 April, the plants grown from plasma-treated seeds were on average 21% taller than the control (Figure 2a). The difference was reduced to 8%, still in favor of plasma treatment, in case of the seeds sown on the intermediate date, while for the latest sowing period the plants’ heights were similar. Improved growth uniformity was also observed for the plasma-treated samples.

Similar behavior was observed with respect to the number of leaves (Figure 2b). Regarding the capitulum diameter (Figure 2c), the largest positive influence of plasma was also seen for the earliest sowing period, when the difference as compared to control plants was of 18%, while for the late-sown seeds, a considerable detrimental effect was noticed.

Figure 3 shows the results on the sunflower plants at harvest, i.e., on 30 September 2021, with regard to the yield elements (capitulum diameter, number of seeds per capitulum and the weight of 1000 seeds). The yield was calculated from these measurements and is shown in Figure 3d.

The data followed a similar trend as observed during plant growth: the positive effect induced by plasma treatment on seeds sown during the optimal period was clearly visible, as well as the detrimental effect of plasma for late-sown seeds. The largest improvement was noticed for the earliest sowing date. In this case, plasma-treated seeds produced plants with 12% larger capitulum diameter (Figure 3a), 23% higher number of seeds per capitulum (Figure 3b) and almost 9% heavier seeds (Figure 3c), on average, as compared to the control. The differences in capitulum size and number of seeds per plant are statistically significant. The resulting mean yield of plasma-treated sunflower seeds was 2319 kg/ha, i.e., over 30% higher than for untreated seeds (Figure 3d).

The variation as compared to control plants was smaller, but still statistically significant, in the case of seeds sown on the intermediate date (26 April). The best results in terms of yield were obtained for this sowing period: the mean yield was 3807 kg/ha, 17% higher than for control samples. For the late-sown seeds (7 May), the effect was reversed, and the results obtained for plasma-treated seeds were significantly worse than for untreated plants. The reasons behind this behavior are still to be determined; they may be related to changes in hormone balance and the subsequent influence of climatic conditions, but this was beyond the purpose of the present paper.

The effect of storage time (the time between plasma treatment and seed sowing) was investigated for a period of two weeks. Sunflower seeds treated by plasma on 7 April (denoted P1 in the next graphs) and on 21 April (P2), were sown on 26 April, and the plants were observed during growth and measured at harvest. Figure 4 presents the comparison of the main results for P1 and P2 (capitulum diameter, number of seeds per capitulum and yield), together with the data for the control samples.

Differences of a few percent were detected between seeds treated by plasma on different dates (P1–P2), in favor of sowing immediately after treatment. However, these variations of the number of seeds per capitulum and of the calculated yield are not statistically significant. The yield for the P1 plants remained considerably higher (14%) as compared to the control. Therefore, it can be concluded that the plasma effect was preserved for two weeks of seed storage; thus, aging does not occur within this time range.

## 3. Discussion

Plant response to non-thermal plasma treatment has often been evaluated by the effect on germination and early seedling growth, as shown in the reviews [1,3,5] and the references cited therein. Many publications reported stimulating effects of plasma on a large variety of plant species, consisting of increased germination and seedling growth; however, less frequently, neutral or even negative effects were also observed. The results largely depend on the experimental parameters, making their optimization crucial in order to obtain positive effects. In addition, various plant species respond differently, sometimes even oppositely, to the same treatment, augmenting the complexity of experiments’ interpretation.

In the present experiments, the germination percentage of sunflower seeds was not improved as a result of plasma treatment; however, it was already high for the control seeds, therefore, no significant enhancement was expected. Zukiene et al. [9] also did not observe considerable differences between the germination curves of untreated sunflower seeds and DBD plasma-treated samples for in vitro germination, while the germination in substrate was negatively affected from the viewpoint of kinetics. The authors found that plasma induces significant changes in the content of seed phytohormones, and these changes strongly depend on the treatment time. These modifications were correlated to the considerable differences in the morphometric parameters, which became visible only after a longer time, i.e., after 30 days of growth in substrate. Thus, a significant increase in root length was noticed for the optimum treatment time, while the stem length was little affected by the plasma; unlike our experiments, where, on the contrary, there was no influence on the roots, but the shoots of plasma-treated samples were considerably longer and heavier.

In other studies, the differences between the control and treated seeds became visible sooner, so the plants were monitored only for several days [6,7]. Sarapirom and Yu [7] suggested that the plasma only acted on the coat of the sunflower seeds, producing modifications in its surface morphology which led to higher water uptake, thus, contributing to the observed improvement in germination and plant growth. This hypothesis, based on the authors’ comparison between peeled and unpeeled seeds exposed to plasma treatment, is, however, debatable, as intimate changes at the molecular level were experimentally proved in other studies [10,11]. The effect of wettability increase cannot be excluded, but much more complex action mechanisms of the plasma are discussed in the literature, including modifications in the content and balance of phytohormones, effects on gene expression, activation of the energy metabolism, etc. [4,5,10,11], which are not entirely understood and should represent the subject of further research.

In a field test, Tamošiūnė et al. [12] observed that plasma treatment stimulated the growth of sunflower lateral organs, and attributed this behavior to the increase in water uptake and/or direct root signaling. The authors associated the enhanced water supply to root growth stimulation caused by changes in the microbial population of the roots observed after plasma treatment. A similar remark with respect to the number of leaves per mature plant applies to the present work. For the optimum sowing period, the number of leaves was significantly higher for the treated seeds as compared to the control seeds. However, in our case, the roots of seedlings were not affected by plasma exposure and we have no information on the roots of mature plants, so our data cannot confirm the above-mentioned hypothesis. Another similarity to the work of Tamošiūnė et al. [12] is the increased capitulum size obtained in our experiments; however, in our case the plasma stimulated stem growth, unlike the results of [12] where height was not affected by the treatment. Overall, this comparison suggests yet again that even for the same plant and the same type of electrical discharge (atmospheric-pressure DBD, in this case), many other factors influence the development of a treated plant, such as the plasma physico-chemical properties, the plant variety, the soil, the climate and so on.

## 4. Materials and Methods

The plasma was generated in a coaxial DBD reactor with a glass tube as dielectric, operated in air at atmospheric pressure. Air flows through the plasma reactor with a rate of 100 mL/min, regulated by a mass flow controller. The inner electrode, a brass rod of diameter Φ_inn_ = 21.7 mm, was connected to high voltage. The outer electrode, an aluminum tape glued on the outside of the glass tube, with a diameter Φ_out_ = 34 mm and length L = 230 mm, was grounded. Approximately 30 g of sunflower seeds were packed inside the reactor. The electrical discharge was operated with sinusoidal voltage of amplitude V_m_ = 16 kV and frequency ν = 50 Hz. The average power dissipated in the discharge was calculated from the voltage-charge plots according to the Lissajous method and had a value of 5 W in all experiments. 

In this work, sunflower seeds of the semi-early hybrid P64LE99 (Pioneer^®^, Corteva Agriscience, Afumati, Ilfov county, Romania), produced in 2019, which were not subjected to any chemical treatment, were exposed to the plasma for 10 min, and plant growth was monitored both under controlled conditions and in the field. This hybrid is characterized by a very high and stable yield potential and a good plasticity and adaptability [14]. The seeds tested in the laboratory, at the Institute for Plant Protection in Bucharest, were incubated for seven days in Petri dishes, on filter paper, at 22 °C temperature, 60% relative humidity and light/darkness program 8/16 h. As a light source, eight Osram L 36W/77 Fluora fluorescent tubes (Ledvance GmbH, Augsburg, Germany) of 1400 lm and 1.2 m length were used. Six replicates of 15 seeds each and one of 10 seeds were used for the two variants: untreated (control) and plasma-treated samples. They were randomly arranged in order to provide equal exposure to light. The germinated seeds were counted on the 2nd, 3rd and 4th day, and the radicle length was measured on the 4th and 6th day. Subsequently, the germinated seeds were transferred into trays with universal organic substrate (commercially available), and plant growth was observed for three more weeks, also using four replicates. Stem length (from the soil up to the cotyledons) was monitored every 2–3 days during the interval between the 10th and the 24th day, the cotyledons’ dimensions were measured on the 10th day, and the seedlings length and weight (fresh and dry) on the 30th day.

The field experiment took place at the Moara Domnească Didactic Farm (44°30′ N, 26°13′ E, 90 m altitude), Ilfov County, in the south-eastern area of Romania. A schematic drawing of the experiment is shown in Figure 5. The previous year the field had been uncultivated after a stone orchard was deforested.

Plasma-treated and control seeds were sown on three sowing dates in 2021: 12 April, 26 April and 7 May, in order to investigate the effect of the sowing period, as well as the influence of storage time. On each of these dates, approximately 100 g of seeds were sown on eight rows of 32 m length, with a distance between adjacent rows of 70 cm. This corresponds to a density of approximately 55,000 plants per ha, and an area of 179.2 m^2^, divided into three replications of a 59.7 m^2^ each, i.e., 8 rows of 10.6 m length. After about two months (i.e., on 29 June–2 July 2021), plant height, number of leaves and capitulum size were measured. The plants were harvested on 30 September 2021 and the capitulum size and weight were measured, as well as the number and mass of seeds per capitulum. Using these data, the yield was calculated and adjusted by 9% standardized moisture content of the seeds [13].

The data were presented as statistical graphs, which allow the complete representation of the useful information for each data set, as shown in the schematic drawing in Figure 6.

The results were analyzed using One-Way ANOVA function in Origin 6.1 software: the means of various parameters were compared for the control and treated samples, and the difference was considered statistically significant at *p* ≤ 0.05.

## 5. Conclusions

As a first conclusion of this study, it can be stated that germination tests under controlled conditions and observation of plant early growth in the laboratory over a short period of a few days are not sufficient to predict the later evolution of plants. Measurements made during the first week of plant growth showed significantly shorter radicles for the plasma-treated seeds as compared to untreated ones. At this stage there was no indication of a favorable outcome of plasma exposure. Subsequent observation during three more weeks revealed, however, positive effects: namely, considerably longer seedlings and more robust plants developed from the treated seeds.

The beneficial effect of plasma treatment was also visible on mature plants cultivated in the field, however, in our case, an influence from the sowing period was observed. The treated seeds sown in April produced taller plants with larger capitulum, increased number of seeds per capitulum and mass per thousand seeds, resulting in the substantial improvement of crop yield. The difference was more important for the earliest sowing date investigated, where the enhancement in sunflower yield reached 30%. On the contrary, in case of late sowing, the treatment had a detrimental effect. This may have been caused by the weather conditions, mainly precipitation, which greatly influences plant development. It was found that the plasma effect lasted for at least two weeks of seed storage, which is useful in case sowing immediately after treatment is prevented by unfavorable climatic conditions. The results obtained in this work are promising; however, the field experiments should be repeated for several consecutive years to confirm the conclusions.

## Figures and Tables

**Figure 1 plants-12-00794-f001:**
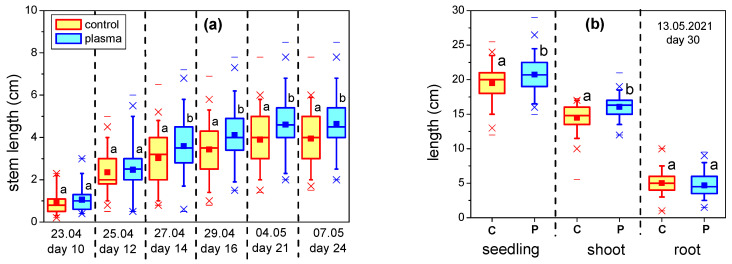
(**a**) Stem lengths (from the soil up to the cotyledons) of plasma-treated and control seeds grown in organic substrate; (**b**) Lengths of roots, shoots and entire seedling after 30 days of incubation; (**c**) Fresh weight and (**d**) Dry weight of the whole plant and, separately, of the shoot and leaves and of the root on the 30th day (C—control/untreated seeds, P—plasma-treated seeds). Different letters indicate statistically significant differences between mean values (*p* < 0.05).

**Figure 2 plants-12-00794-f002:**
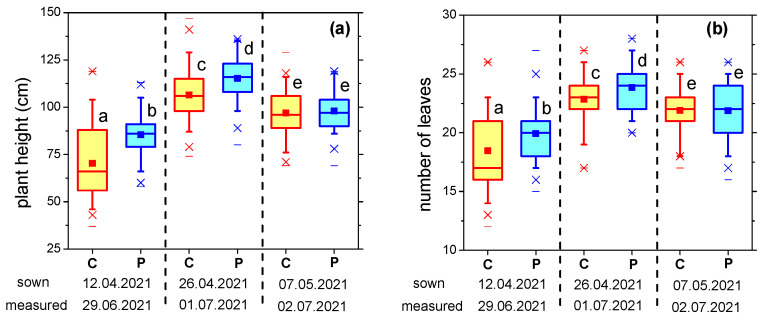
(**a**) Plant height; (**b**) Number of leaves; (**c**) Capitulum diameter of sunflower plants grown from plasma-treated and control seeds, at approximately 2–2.5 months from sowing (C—control/untreated seeds, P—plasma-treated seeds). Different letters indicate statistically significant differences between mean values (*p* < 0.05).

**Figure 3 plants-12-00794-f003:**
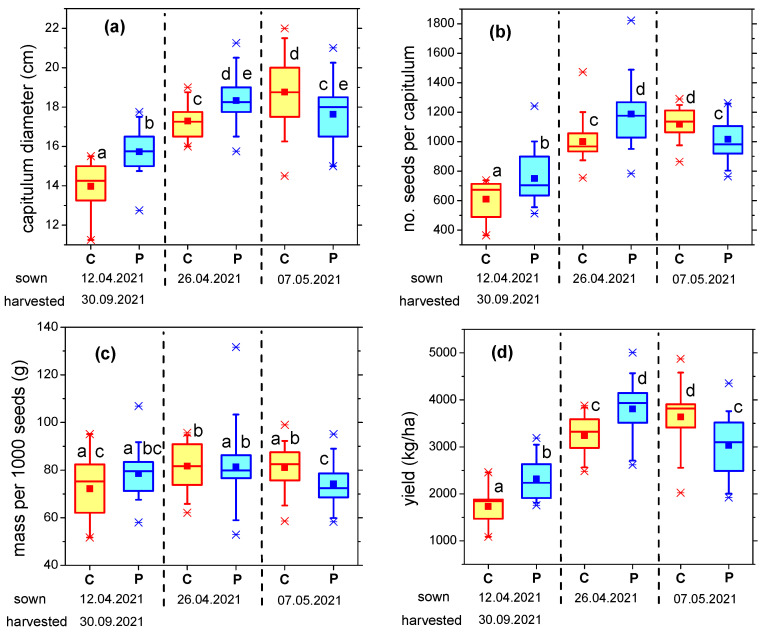
(**a**) Capitulum diameter; (**b**) Number of seeds per capitulum; (**c**) Mass per 1000 seeds; (**d**) Yield, for sunflower plants grown from plasma-treated and control seeds (C—control/untreated seeds, P—plasma-treated seeds). Different letters indicate statistically significant differences between mean values (*p* < 0.05).

**Figure 4 plants-12-00794-f004:**
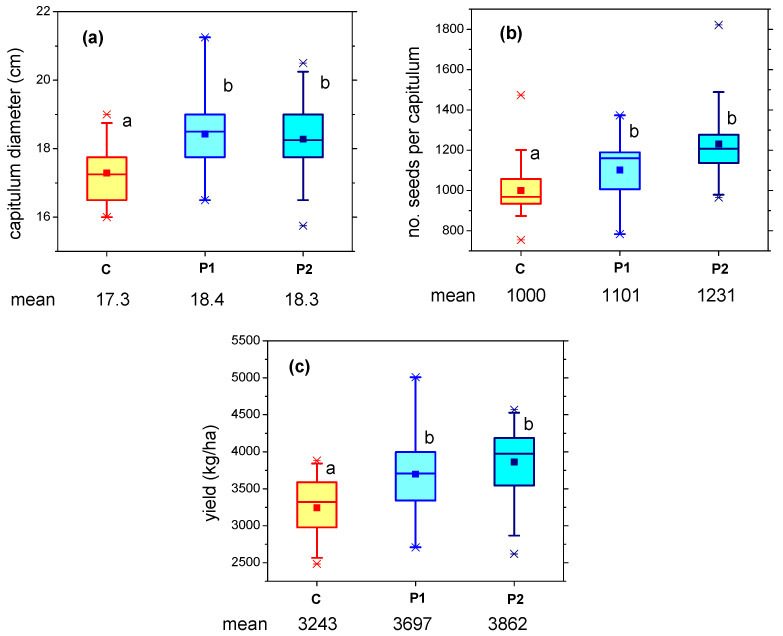
(**a**) Capitulum diameter; (**b**) Number of seeds per capitulum; (**c**) Yield, for sunflower plants grown from untreated/control seeds (C) and seeds treated by plasma on 7 April 2021 (P1) and on 21 April 2021 (P2), sown on 26 April 2021 and harvested on 30 September 2021. Different letters indicate statistically significant differences between mean values (*p* < 0.05).

**Figure 5 plants-12-00794-f005:**
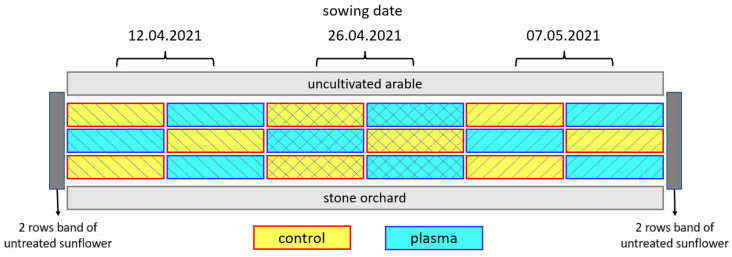
Schematic drawing of the field experiment.

**Figure 6 plants-12-00794-f006:**
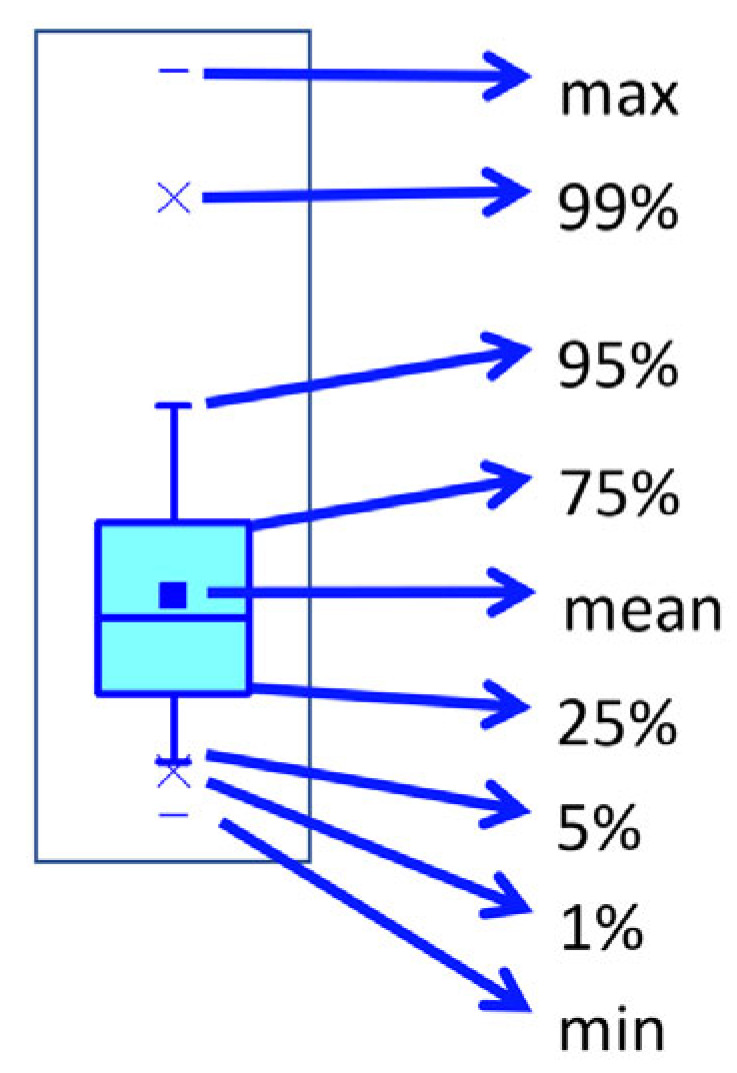
Graphic representation of the results for each data set throughout this paper.

## Data Availability

The data presented in this study are available on request from the corresponding author.

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
