# Peer review of "Positive Effect Induced by Plasma Treatment of Seeds on the Agricultural Performance of Sunflower"

_plants, 2023, doi:10.3390/plants12040794_

Round 1
Reviewer 1 Report
Dear authors,
Many thanks for your contribution. It shows that plasma treatment of sunflower seeds can have positive effect on the grove as well as final crop. Unfortunately, there is impossible to agree with your conclusions without proper description of all comments given mainly under point 6. Thus, however results seem be interesting, paper could not be considered for the publication. I’m understanding you that for physicists this description seems be OK but for others current description is fully insufficient even if we consider this as a pilot experiment.
1. What gas and pressure was used for the seeds treatment?
2. Were seeds moved (mixed) during the treatment to obtain homogeneous surface treatment?
3. Used seeds there chemically treated before or un-pre-treated seeds were used?
4. What kind of substrate was used? This can significantly affect the obtained results.
5. What light source was used? And was illumination uniform? To avoid these effects, replications of the same treatment must be spread randomly.
6. Results of field experiment form one season are not sufficient for proper evaluation because results differ season to season. Commonly, at least 3 consequent years are necessary to get relevant results. Scheme of thew field experiment is fully missing. Commonly, at lest 3-5 replication randomly spread over the experimental field are necessary to minimize influence of the soil inhomogeneity. Additionally, what was grown on experimental field before and was whole field area uniform during precedent years? And what was in surrounding? Without this information, it is impossible to make any serious conclusions. How many plants were grown?
7. The number of related references is very low compared to the contemporary citations inflation.
Reviewer 2 Report
Dear Editor
Thank you very much for your invitation to review this manuscript:
Positive effect induced by plasma treatment of seeds on the agricultural performance of sunflowerComments
The work is reported on the Positive effect induced by plasma treatment of seeds on the agricultural performance of sunflower.
I read the wark carefully, it introduces very important information but the discussion is very poor and should be separated from the results. Also, the authors should follow the comments in pdf version in the following lines:
Line 71
Line 119
Line 131
Line 132
Line 157
Line 190
Line 192
Line 198
Line 199
Line 231
Line 232
Line 235
Line 294 and all the references
Please see the pdf version
Best regards

Round 2
Reviewer 1 Report
Dear authors.
Many thanks for the improvements and explanations. Now it is much clearer than before. I'm recommentding to modify conclusions, only. Do not be so strong because there is impossible to make strong conclusions from one year experiment in the field conditions. Typically, the main effect is based on weather conditions, mainly precipitation. This is not presented in the paper. Thus, you can compare seeding at the same time, this is correct, but results reflets mainly weather, not simply date of seeding. I hope that experiment will be repeated in at least additional two years to get more relevant conclusions.
Author Response
We thank the reviewer for the useful comments. We made the required changes in the conclusions to acknowledge the essential influence of weather conditions, mainly precipitation, on plant development. Repeating the experiments for a few years in a row is also our goal, in order to confirm the conclusions.
Reviewer 2 Report
I accept the manuscript in present form
Author Response
We thank the reviewer for considering our manuscript suitable for publication.